# Phosphoproteomic Analysis of Rat Neutrophils Shows the Effect of Intestinal Ischemia/Reperfusion and Preconditioning on Kinases and Phosphatases

**DOI:** 10.3390/ijms21165799

**Published:** 2020-08-13

**Authors:** Muhammad Tahir, Samina Arshid, Belchor Fontes, Mariana S. Castro, Simone Sidoli, Veit Schwämmle, Isabelle S. Luz, Peter Roepstorff, Wagner Fontes

**Affiliations:** 1Laboratory of Protein Chemistry and Biochemistry, Department of Cell Biology, University of Brasilia, Brasilia 70910-900, Brazil; tahir.bio@gmail.com (M.T.); saminatahir83@yahoo.com (S.A.); mscastro69@gmail.com (M.S.C.); isabelle.sluz@gmail.com (I.S.L.); 2Department of Biochemistry and Molecular Biology, University of Southern Denmark, DK-5230 Odense M, Denmark; simone.sidoli@gmail.com (S.S.); veits@bmb.sdu.dk (V.S.); roe@bmb.sdu.dk (P.R.); 3Laboratory of Surgical Physiopathology (LIM-62), Faculty of Medicine, University of São Paulo, São Paulo 01246903, Brazil; belchor@uol.com.br; 4Department of Biochemistry, Albert Einstein College of Medicine, Bronx, NY 10461, USA

**Keywords:** systemic inflammatory response, ischemia and reperfusion, preconditioning, neutrophils, phosphorylation, proteome, kinases, phosphatases

## Abstract

Intestinal ischemia reperfusion injury (iIRI) is a severe clinical condition presenting high morbidity and mortality worldwide. Some of the systemic consequences of IRI can be prevented by applying ischemic preconditioning (IPC), a series of short ischemia/reperfusion events preceding the major ischemia. Although neutrophils are key players in the pathophysiology of ischemic injuries, neither the dysregulation presented by these cells in iIRI nor the protective effect of iIPC have their regulation mechanisms fully understood. Protein phosphorylation, as well as the regulation of the respective phosphatases and kinases are responsible for regulating a large number of cellular functions in the inflammatory response. Moreover, in previous work we found hydrolases and transferases to be modulated in iIR and iIPC, suggesting the possible involvement of phosphatases and kinases in the process. Therefore, in the present study, we analyzed the phosphoproteome of neutrophils from rats submitted to mesenteric ischemia and reperfusion, either submitted or not to IPC, compared to quiescent controls and sham laparotomy. Proteomic analysis was performed by multi-step enrichment of phosphopeptides, isobaric labeling, and LC-MS/MS analysis. Bioinformatics was used to determine phosphosite and phosphopeptide abundance and clustering, as well as kinases and phosphatases sites and domains. We found that most of the phosphorylation-regulated proteins are involved in apoptosis and migration, and most of the regulatory kinases belong to CAMK and CMGC families. An interesting finding revealed groups of proteins that are modulated by iIR, but such modulation can be prevented by iIPC. Among the regulated proteins related to the iIPC protective effect, Vamp8 and Inpp5d/Ship are discussed as possible candidates for control of the iIR damage.

## 1. Introduction

Intestinal ischemia reperfusion (iIR) injury can be caused by various clinical conditions triggering a severe clinical syndrome and even death. These conditions include acute mesenteric ischemia, incarcerated hernia, small intestine transplantation, and intestinal obstruction [1,2]. Moreover, mesenteric ischemia may also be a consequence of hemorrhagic, cardiogenic, or septic shock [3]. The overall mortality rate due to acute mesenteric ischemia is 60% to 80% with increasing incidences over time. The iIR injury not only results in local acute inflammatory response but also pulmonary injury due to the infiltrating neutrophils and systemic inflammatory changes leading to development of multiple organ failure (MOF) [4,5,6,7,8]. Various studies have reported the key role of neutrophils in the pathophysiology of iIR as a result of leukocytes and endothelial cells (ECs) interactions in the intestinal mucosa [9]. The clinical outcome from iIR conditions is related to an interplay among neutrophils and many other cell types in the inflammatory tissue microenvironment, such as the homeostatic balance between PMN and mucosal cells in HIF regulation [10], or the correlation of neutrophil-to-lymphocyte ratio with mesenteric thrombosis [11]. Depletion of neutrophils from blood before reperfusion in human small bowel decreases the effect of IR [12]. Neutrophils contribute to the tissue damage by proteolytic enzyme secretion from cytoplasmic granules and production of free radicals by respiratory burst, causing damage to the microcirculation and prolonging ischemia [13,14,15,16]. 

Various strategies including pharmacological approaches were tested to reduce neutrophil infiltration to the tissues that resulted in reduced ischemia and reperfusion injury [6,17]. Neutrophil blocking or depletion results in less organ failure in the pro-inflammatory phase; however, increased organ failure incidence can be seen by severe infection in the anti-inflammatory phase. Instead of shutting down an important defense mechanism, it seems more favorable to regulate the neutrophil behavior. A phenomenon called ischemic preconditioning (IPC) was reported about 30 years ago that protects the organs against IR injury [18]. The IPC is a phenomenon triggered by short episodes of ischemia and reperfusion before a long ischemia and reperfusion. The beneficial role of IPC is clearly evident from the literature, and it has been tested in different organs including intestine, brain, spinal cord, kidney, lung, liver, and human heart [19,20,21,22,23,24,25,26]. There are some studies explaining the changes in the neutrophil behavior after IPC. Ischemia reperfusion causes systemic neutrophil activation with induced CD11b expression and neutrophil-platelet complex formation. Consequently, profound endothelial dysfunction can be observed; however, the IPC attenuates both the effects [27], even though it can be considered as a first stimulation of the inflammatory response [28,29]. The protein abundance profile of the neutrophil exposed to iIRI changes drastically, presenting alterations in phagocytosis and ROS production enzymes [30]. Some of these regulated proteins have their abundance profile kept at control levels if the IPC is applied, and most of the enzymes presenting this profile are hydrolases or transferases [31]. Less frequent, but still significant abundance changes were found for proteins related to the protein biosynthesis and proteasome degradation processes in mild surgical trauma [32]. Since protein phosphatases are a subclass of hydrolases and protein kinases are a subclass of transferases, the analysis of protein phosphorylation was developed as a follow-up study from our previous work.

Protein phosphorylation is an essential and one of the most common post-translational modifications (PTM) affecting cell cycle progression, gene expression, signal transduction, and other cellular activities [33]. Protein kinases, over 500 members, phosphorylate proteins at serine (S), threonine (T), and tyrosine (Y) residues. It has been predicted that kinases encompass 1.7% of the human genome [34]. The estimated relative abundance of S/T/Y phosphorylation found in human proteome is 86.4%/11.8%/1.8%, respectively [35]. Protein phosphatases catalyze the enzymatic removal of the phosphate group from these residues (S/T/Y) and return the proteins to their non-phosphorylated state [36]. In addition to the protein function regulation due to conformational changes, phosphorylation sometimes disrupts surfaces for protein–substrate interactions and vice versa [37,38,39]. Kinases and phosphatases regulate the complex dynamics of protein phosphorylation that can be disrupted in various diseases including diabetes, cancer, neurodegenerative, and autoimmune diseases [40,41]. 

In that sense, the main objective of the present work was to analyze the phosphoproteome of neutrophils from rats submitted to mesenteric ischemia and reperfusion, either submitted or not to IPC, compared to quiescent controls and sham laparotomy. In such dataset, we sought for evidences of phosphorylation-based regulation of events related to the iIRI and its prevention by iIPC. Among our findings, it is interesting to mention that most of the phosphorylation-regulated proteins are involved in neutrophil apoptosis, migration, and degranulation, and most of the regulatory kinases belong to CAMK and CMGC families. Some relevant regulatory proteins discussed are PKCδ and NDR1/Stk38, and some of them, such as Inpp5d/Ship and Vamp8, present an abundance profile of phosphopeptides that are compatible with modulation by iIR and prevention of such modulation by iIPC.

## 2. Results

### 2.1. Phosphoproteome Analysis of Rat Neutrophils

A total of 40 rats were randomly allocated into the four experimental groups. These groups included the control group (Ctrl, without any surgery), sham laparotomy group (Lap, without clamping of any artery but received the same surgical procedure), intestinal ischemia and reperfusion group (iIR, had 45 min of superior mesenteric artery occlusion (SMAO) followed by 120 min of reperfusion), and intestinal ischemic preconditioning (iIPC, subjected to 10 min of SMAO followed by 10 min of reperfusion immediately before 45 min of ischemia and 120 min of reperfusion). About 7–10 mL of blood was collected from the heart after the surgical procedures, and rat neutrophils from the four different groups (Ctrl, Lap, iIR, and iIPC) were isolated and processed for protein extraction. After the tryptic digestion, the peptides were iTRAQ labeled and multiplexed, followed by the enrichment for phosphorylated peptides. The purified peptides were HILIC fractionated and analyzed by mass spectrometry (MS) (Figure 1). The MS analysis of the peptides resulted in the identification of a total of 2151 proteins (Appendix A), out of which 549 were phosphorylated proteins (Figure 2A). For protein identification, a standard criterion of two peptides per protein was used, whereas the site localization of the phosphorylation was accepted with phosphoRS score of > 95%. The Venn diagram shown in Figure 2B,C represents regulated phosphorylated proteins and peptides. Unique and overlapping phosphosites for serine, threonine, and tyrosine (S/T/Y) were observed with phosphorylation abundance order of S > T > Y. The number of phosphorylated peptides with Sph, Tph, and Yph showed a pattern distribution of 75%, 22%, and 3%, respectively (Figure 2D), and a similar pattern was followed by the phosphosites (Figure 2E). The most common phosphorylated residue was serine (Sph) in all of the phosphorylated proteins followed by threonine (Tph) and tyrosine (Yph) phosphorylation at 60%, 33%, and 7%, respectively (Figure 2F).

A cluster-based analysis of the total proteins and phosphopeptides was performed using an in-house developed R script as described in the methodology section (Appendix A). A total of six different clusters resulted after analysis based on the abundance profiles in the Ctrl, Lap, iIR, and iIPC groups (Figure 3A). Clusters 1 and 6 show a progressive up- and down-regulation of the phosphorylation profiles, respectively. Such a profile could be related to modifications progressively induced by long traumatic surgical procedures. Cluster 2 represents down-regulation in iIPC compared to Lap and iIR, reflecting the possible effects of iIPC alone, regardless of previous stimulation. However, clusters 3 and 5 show a pronounced effect with sharp down- or up-regulation (respectively) in iIR, and an opposite regulation in iIPC, showing proteins that might be related to the protective effect of iIPC. Cluster 4 represented a sharp decline in the expression profile in Lap. In cluster 1, 378 phosphopeptides from 188 proteins were grouped; cluster 2 carried 621 phosphopeptides from 229 proteins; cluster 3 had 541 phosphopeptides from 232 proteins; 565 phosphopeptides from 224 proteins were in cluster 4; 754 phosphopeptides from 249 proteins were in cluster 5; cluster 6 had 651 phosphopeptides from 251 proteins (Figure 3B). A different number of Sph, Tph, and Yph sites were seen across the clusters. Most of the phosphorylation regulation was observed in cluster 5, followed by clusters 6, 2, and 3.

### 2.2. Kinases and Phosphatases Phosphorylation in Rat Neutrophils

A total of 12 protein kinases and 4 phosphatases were identified in our analysis with significantly differential phosphorylation in at least one of the residues. Most of them contain phosphorylation in catalytic domains of the proteins. Five of the kinases identified were found with significant regulation in iIR vs. Ctrl, and two kinases were significantly regulated (FDR < 0.05) in iIR vs. iIPC. Significant differences in iIR vs. Ctrl, but not to the Lap group, might indicate proteins that were affected by tissue damage, regardless of the damage intensity, or gradually from Ctrl to Lap to iIR, but only showing significance between Ctrl and iIR. While significant differences in iIR vs. iIPC might indicate proteins involved in the protective effect of iIPC. The identified kinases and phosphatases with cluster-wise abundance profile, number and position of phosphosites, and their presence in domain region are given in Table 1. Two tyrosine-protein kinases important for the neutrophil function, fgr and syk, were found with significant regulation in 7 peptides and belonged to cluster 2. The most important phosphorylation was found in the catalytic domain at tyr400. The phosphorylation was higher in iIR compared to Ctrl, suggesting a decrease in domain phosphorylation in iIPC as well. The syk protein was found with one phosphorylation regulation in iIPC compared to Ctrl in our study and not in any domain region. Another kinase, beta-adrenergic receptor kinase 1, was found with significantly higher phosphorylation at S670 in iIR compared to Ctrl. The peptide was assigned to cluster 2, and a phosphorylation site was found in the PH domain region. Another important kinase, protein kinase c delta, was found with significant regulation in phosphorylation at position S643 and S645. The peptide was assigned to cluster 6 (lower in iIR and iIPC compared to Ctrl and Lap), and phosphorylation sites were in the catalytic domain of the protein. Another kinase with regulated phosphorylation in the catalytic domain at position T452 identified was serine/threonine-protein kinase 38. The phosphorylation was up-regulated in iIR compared to iIPC. 

There are some phosphatases identified in our proteomic analysis of the rat neutrophils with significant regulation in phosphorylation in the domain regions. Tyrosine-protein phosphatase non-receptor type 6 was found with 2 phosphorylation sites with up-regulation in iIR compared to iIPC and assigned to cluster 2. The up-regulation in phosphorylation was found at Ser12 and in N-SH2 domain. Another phosphatidylinositol, 3,4,5-triphosphate 5-phosphatase 1, was identified with four up-regulated phosphorylation sites in iIR compared to iIPC and assigned to cluster 3. The phosphorylation at T519 was found in a catalytic region called SH2 domain and putative catalytic site, putative active site, putative Mg-binding site, and also in putative PI/IP-binding sites of this phosphatase.

A number of other kinases and phosphatases with significant regulation in their phosphorylation were also identified, but their phosphorylation was not found to be in any of the domain regions. These kinases and phosphatases are listed in Table 1.

### 2.3. Motif-x Enrichment Analysis of Phosphopeptides

The Motif X algorithm was used for the analysis of the sequence motif enrichment of phosphorylation sites (Table 2). The amino acids window of 17 and *Rattus norvegicus* background database were selected as parameters. Proteins with highly significant phosphorylation site at serine in their motifs showed more abundance (138) compared to the highly phosphorylated threonine residue in their motifs (8). The logos observed in Motif X are shown in the Appendix A. Some of the proteins were observed to have both serine and threonine in their motifs with high regulation (Figure 4A). 

After the motif-x analysis, distribution of the enriched serine and threonine motifs were checked across the six clusters (Figure 4B). As presented, the abundance of S.S motif was somewhat homogeneous through clusters 2, 3, 5, and 6, whereas SP motif was more prominent in these clusters. Comparatively, clusters 1 and 4 have the least motif containing phosphopeptides. The phosphopeptides with threonine motifs were prominently found in cluster 2, 3, and 5, whereas no enrichment was found for the tyrosine motifs, as the tyrosine containing peptides were low in number compared to serine- and threonine-containing phosphopeptides. 

### 2.4. Prediction of Kinases Responsible for Regulated Phosphorylation in Domain Regions

A group-based prediction system (GPS) software package [42] was used for the phosphoproteome data. A prediction analysis was performed for the in vivo site-specific kinase-substrate relation for the proteins with regulated phosphorylation in the domain regions Table 3. The distribution of all the predicted kinase families responsible for the phosphorylation event in the serine and threonine motifs was grouped in pie charts (Figure 5). It is clear from Figure 5A that the kinase families with higher percentage for phosphorylation of serine motifs are CAMK, CMGC, and AGC, whereas for threonine motifs, they belong to CMGC among other families (Figure 5B). A list of commercially available inhibitors for these predicted kinases is given in the Appendix A. 

### 2.5. Pathways Analysis of the Proteins with Significantly Regulated Phosphorylation

To analyze the effect of the iIPC on the regulated pathways in rat neutrophil proteome, the phosphoproteome data was searched against KEGG pathways database. The most significantly regulated pathways observed included spliceosome, chemokine signaling pathway, Fc-gamma R-mediated phagocytosis, and focal adhesion. Each pathway involved a different number of the phosphorylated proteins (Table 4).

## 3. Discussion

Protein phosphorylation is an essential post-translation modification, and the eukaryotic cells rely extensively on it for their basic cellular processes like metabolism, motility, organelle trafficking, muscle contraction, immunity, learning, and memory [34,43]. Signal transduction depends on the phosphorylation of the hydroxyl group on the side chain of serine, threonine, and tyrosine [44,45]. Various studies have documented different ratios of S/T/Y phosphorylation in different cell lines; Macek et al. reported a ratio of 70:20:10 in *B. subtilis*, and in another study, the same group found 86:12:2 for S/T/Y in human cell culture [35,46]. The serine–threonine–tyrosine ratio in our study for the rat neutrophil phosphoproteome was 75:22:3 (Figure 2B), whereas the phosphoproteomics analysis ratio of 14 different rat organs and tissues by Lundby et al. was 88.1:11.4:1.5 for S/T/Y, respectively [47]. In the present study, we analyzed the phosphoproteome of the rat neutrophil. For the relative quantification, an iTRAQ-based approach was adopted. Although presenting some limitations [48], the method is very robust for the large number of biological replicates and conditions, especially when metabolic labeling is not feasible due to the use of non-dividing cells. This labelling approach, in addition to others like TMT, allows not only the reproducible enrichment for phosphorylated peptides but also for other post-translational modifications in parallel in one experiment [49,50]. 

Phosphorylation events are regulated by protein kinases and phosphatases. The protein kinases play a vital role in cell signaling cascades and make up to 2% of the genome [43,51,52]. It is clear now that short linear motifs around the site of phosphorylation provide primary specificity for kinase substrate recognition [53]. The cluster 4 represents the effect of laparotomy on neutrophil phosphoproteome (Figure 3A). During our analysis, the laparotomy group did not have a statistically significant effect on phosphoproteome of the kinases and phosphatases, compared to the iIR and iIPC. There are very few proteins with significantly higher phosphorylation regulation in the laparotomy group compared to the control group, like the up-regulation of 60S acidic ribosomal protein P2 (P02401) and the down-regulation of 26S proteasome non-ATPase regulatory subunit 1 (O88761), in agreement with previous findings of higher rates of mRNA processing and protein synthesis for the neutrophil after surgical trauma [32]. 

### 3.1. Phosphorylation Events Related to the Ischemia-Reperfusion Injury

A considerable number of regulated phosphosites was associated to differences between the iIR and the Lap or the Ctrl groups, suggesting possible molecular mechanisms related to the ischemia-reperfusion injury. An important tyrosine-protein kinase, Fgr, a member of Scr family, was found with seven significantly regulated phosphorylation sites, one of them (tyr400), found in the catalytic domain, was assigned to cluster 2 with up-regulation after ischemia, compared to Ctrl. Phosphorylation in the activation loop of the kinase domain at this position switches the kinase to its active conformation and regulates migratory ability and chemokine secretion of the neutrophil [54,55,56]. The up-regulation of the phosphorylation in the catalytic domain in iIR suggests Fgr activation in neutrophils regulating its migration and chemokine secretion after iIR. Interestingly, in our analysis, five regulated phosphopeptides were clustered in cluster 5 containing three motifs, SP, S.S, and S…….S (Table 1 and Table 2). This makes the Fgr kinase a strong candidate for modulation of the neutrophil function with multiple catalytic functions contributed by the distant motif sequences of the protein [57]. Proto-oncogene tyrosine-protein kinase, Src, is responsible for the specific phosphorylation of tyrosine residues in other proteins [58]. In our search analysis, it is the predicted kinase with the highest score that might be a possible candidate responsible for the Fgr phosphorylation at Y400 (Table 3). Another relevant kinase for the possible Y400 phosphorylation is tyrosine-protein kinase, Tec, which plays an important role in human neutrophil responses to monosodium urate [59]. The high score of the predicted kinase indicates a high possibility of the residue to be phosphorylated by that kinase. Although some of the predicted kinases like Tec do not have proper inhibitors, others, such as Src, can be further tested by inhibition of the phosphorylation event to study the neutrophil migration and induced damage to the host tissues or organs after ischemia and reperfusion (Appendix A). 

The β-adrenergic receptor kinase-1 (Grk2/P26817) has a C-terminal PH domain that possesses binding specificity for Gβγ subunits and for the membrane phospholipid PIP2 [60,61,62]. We identified a serine residue at S670 with SP motif in the PH domain of this protein with significantly higher phosphorylation in iIR compared to Ctrl. Its higher expression in neutrophils desensitization to chemoattractants was observed in sepsis and in neutrophil treatment with cytokines plus LPS [63]. In our analysis, its regulation in phosphorylation may affect the sensation of neutrophil towards chemoattractant and hence their migration. Among the five possible different kinases responsible for this specific S670 phosphorylation in Grk2, CMGC/MAPK has the highest score and known inhibitors. It has been shown previously that S670 phosphorylation by ERK1/2 kinases impaired strongly Grk2/Gβγ interaction, inhibiting its catalytic activity and kinase translocation towards membrane receptor substrates. However, phosphorylation of the same serine residue by MAPK kinase disrupts the complex allowing Grk2/GIT1 association and favors cell migration preceded by the localized activation of Rac/PAK/MEK/ERK signaling [64]. Our finding shows that Grk2 phosphorylation might be because of MAPK or ERK1/2 that can alter Grk2 catalytic activity in neutrophils after ischemia towards membrane receptor substrates and consequently activation of various cascades via GPCR signaling. 

Protein kinase C delta (PKCδ) is a novel PKC that plays an important role in neutrophil transmigration mediated by IL-1 beta and fMLP by regulating neutrophil adherence and was identified in our analysis with two significantly regulated phosphorylation sites (S643 and S 645) in the catalytic domain in iIPC versus control. [65]. S643 phosphorylation has already been shown to be required for catalytic maturation of PKCδ, whereas in neutrophils the role is not known. Enhanced EC survival was observed after decreased activity of this enzyme by hypoxia [66]. In the present work, two kinases, CAMK/RAD53 and CAMK/CAMK1, were predicted with the highest score for the regulated phosphosites S643 and S645, respectively. These multifunctional CAMKs belong to a family of serine/threonine kinases that is sensitive to intracellular Ca^2+^ and affects various cellular activities like gene expression, cell cycle, differentiation, and ischemic tolerance [67]. 

Serine/threonine-protein kinase (OSR1) and serine/threonine-protein kinase 38 (Stk38) are the next two important kinases we identified with significantly higher phosphorylation in iIR compared to iIPC. The Stk38, also known as Nuclear Dbf2-Related kinase 1 (NDR1), has a regulated phosphosite on threonine residue T452 in the catalytic domain region (Table 1). Selimoglu et al. found that osmotic pressure and oxidative stress result in NDR1 phosphorylation on T444 by MAP4k4 kinase and induce apoptosis [68,69]. The possible kinases for T452 site phosphorylation are tyrosine kinase like (TKL) and PKC, and the exact role of this phosphosite is not known yet. AGC kinase family including PKC mainly regulates different physiological processes important for metabolism, cell growth, proliferation, and survival [70]. In neutrophils, phosphorylation of NDR1 in the catalytic domain can be a good target during ischemia reperfusion that consequently might lower tissue damage, whereas in iIPC, the phosphorylation was significantly lower compared to iIR, validating its protective role. 

Significantly regulated phosphopeptides were identified in some other kinases in our analysis, but the respective phosphosites were not localized in any domain region of the kinases. Serine/threonine kinase N1 (Pkn1) is one of the members of protein kinase C superfamily of serine/threonine kinases and was found with up-regulation in phosphorylation in iIR compared to Ctrl at serine residue S377 with S.S motif. The serine/threonine kinase family is one of the first identified effectors for RhoA-GTPase that exists in integral plasma membrane and cytosolic pool [71]. The spleen tyrosine kinase (Syk) was found with significant down-regulation in S.S motif region at serine S291 residue in iIR compared to Ctrl group during our analysis. It has been found that CD18 integrins downstream signaling is dependent on Syk, and loss of Syk kinase mediates integrin signaling impaired activation of leukocytes resulting in reduced host defense responses [72]. For neutrophils, Syk is an important player and required for phagocytosis, and its inhibition lowers the phagocytosis of IgG coated particles [73]. Some studies show its role in making the actin filament cup during FCγR-mediated phagocytosis [74]. The significant down-regulation of the phosphorylation on S291 site might affect actin filament formation or defense response after iIPC. Another kinase, STE20-like serine/threonine-protein kinase (Slk), found with a significantly regulated phosphosite, exacerbates apoptotic activity and may regulate survival of the cell during repair or injury [75]. There are some studies showing the apoptotic correlation of Slk expression in kidney cells subjected to in vitro ischemia reperfusion injury [76]. However, other findings show its effect on cytoskeletal reorganization as part of anti-apoptotic cascades [77,78]. Oxsr1 is an oxidative-stress responsive protein and belongs to the STE20 family. It regulates the Na+/H+ exchanger activity [79]. In our analysis, significantly up-regulated phosphorylation was found for Oxsr1 at serine residues S324 and S359 in SP motif in iIR surgical group (Table 1), suggesting its role in reducing the production of reactive oxygen species (ROS) and NADPH oxidase inhibition [80]. Pak2 was also identified with two phosphorylated peptides regulated only in iIR compared to Ctrl and Lap, and it has been shown as one of the most regulated proteins for actin cytoskeleton regulation pathway playing a key role in directional migration [31]. 

It has been well documented that phosphatases cause de-phosphorylation of the phosphorylated proteins. Some important protein phosphatases with significant regulation in their phosphorylation have been identified in our analysis (Table 1). Phosphatases have regulatory sequences that flank the catalytic domain and control the activity by interactions at the active sites [81,82]. Protein tyrosine phosphatase receptor type C (Ptprc/CD45) is a phosphatase identified in our analysis with one phosphosite on S1209 significantly regulated in iIR compared to Lap and iIPC groups. It is commonly known as LCA (leukocyte common antigen) and has an important role in neutrophil adhesion, chemotaxis, ROS production, phagocytosis, and bacterial killing [83]. Another phosphatase called tyrosine-protein phosphatase non-receptor type 6 (Ptpn6/Shp-1) was found with an up-regulated phosphosite on S12 in SH2_N-SH2_SHP_like domain and assigned to cluster 2. The N-terminal SH2 domain regulates Shp-1 catalytic activity negatively by binding directly to Shp-1 catalytic domain. Shp-1 also modulates apoptosis signaling cascades in neutrophils [84]. Specific surface immune inhibitory receptors containing immunoreceptor tyrosine-based inhibitory motifs (ITIM) in their cytoplasmic domain mediate inhibitory signaling [85]. Following receptor activation, the Src family kinases phosphorylate a tyrosine residue of the ITIM domain and recruit SH2 domain-containing tyrosine phosphatases (Shp-1). Their subsequent phosphorylation results in the deactivation of tyrosine kinases inhibiting the survival signaling, especially TNF-α, TRAIL, Fas ligand, G-CSF, GM-CSF, or IFN-γ in neutrophils [86]. Shp-1 activation in inhibition of survival signaling leads to longer lifespan found in activated neutrophils after iIR surgical procedures. The identified significant phosphosite regulation on S12 in N-Sh2 domain of Shp-1 can be an important possible candidate for regulating these mechanisms and needs further validation. Different kinases have been reported to bind to Shp-1 through a novel phosphorylation-independent kinase tyrosyl inhibitory motif [87]. We identified significant regulation in phosphosite S12, in the R..S motif having binding possibility for kinases from CAMK and CMGC families (Table 1, Table 2 and Table 3). The role of CAMK and CMGC kinases dependent Ptpn6 has been shown to positively regulate TCR signaling cascades in T cells; however, the role of CAMKs in neutrophils to regulate phosphatases like Ptpn6 has not yet been described and requires further investigations [88]. Phosphatidylinositol 3, 4, 5-triphosphate 5-phosphatase 1 (Inpp5d/Ship) was identified with significant up-regulation in iIR compared to iIPC surgical group. It was assigned to cluster 3 with four regulated phosphosites that were localized in catalytic inositol polyphosphate 5 phosphatase (Inpp5d) domain, SH2 domain, putative active and catalytic sites, and putative Mg and PI/IP binding sites. Ship modulates most of the phosphatidylinositol 5’-phosphatase activity and plays a key role in neutrophils for their efficient cytoskeleton polarization and directional migration. However, spontaneous neutrophil death may result from its activation and consequently down-regulating Akt [89,90]. Ship has been well known for negatively regulating most of the immune and cellular responses like phagocytosis, degranulation, proliferation, cell migration, survival, and sensitivity to chemokines [91,92]. The phosphosite regulation in iIR in cluster 3 might negatively regulate neutrophil dependent inflammatory processes like in acute lung injury resulting in spontaneous death, chemotaxis, and directional migration after iIPC [93]. Therefore, Ship activation can be an important step in protection against iIR, found associated with iIPC. The predicted Tousled-like kinase (Tlk) (Table 3) belongs to a family of serine/threonine kinases that are conserved in plants and animals and have an important role in cell cycle progression by regulating chromatin dynamics [94]. Its proper interaction and regulation for Ship is not well known yet.

### 3.2. Molecular Events Associated to iIPC

Importantly, clusters 3 and 5 show abundance profiles of proteins and phosphopeptides that are profoundly affected by iIR but such effect is not observed in iIPC (Figure 3A), indicating a possible correlation to the protective effect of iIPC. The unsupervised pathways analysis searches for the rat neutrophil phosphorylated proteins with significantly regulated phosphorylation in any of the two conditions from clusters 3 and 5 were performed against KEGG database (Table 4). The most significantly enriched pathway was spliceosome with four proteins, u2af2, srsf2 (SR family), sf3b1, and usp39. In humans, about 200 spliceosomal and splicing-associated proteins regulate alternative splicing (AS) altering cytokine signaling cascades and modulating protein functions [95]. Some of the proteins assigned to spliceosome with regulated phosphosites were u2af2, sf3b1, and usp39. Changes in these proteins were found related to myeloid malignancy [96] (u2af2), clinical monoclonal B-cell lymphocytosis [97] (sf3b1), and prostate cancer [98] (usp39). The srsf2 protein in stimulated T cells was found to regulate CD45 [99]. AS can regulate protein expression in a cell-specific and tissue-specific manner in response to environmental and developmental cues; however, the exact knowledge of spliceosome dependent gene expression regulation in neutrophil is incomplete.

Both the insulin and mTOR-signaling pathways play a pivotal role in a variety of neutrophil functions. Insulin is known as an anabolic hormone that inhibits the degradation of carbohydrates, lipids, and proteins and helps in increasing their synthesis and storage [100,101]. The insulin-signaling pathway maintains glucose homeostasis. It has also been reported that insulin plays a key role in dose-dependent chemotaxis in monocytes and macrophages during wound healing [102]. Another study found an indirect confirmation that priming of neutrophil respiratory burst is a result of crosstalk of signaling pathways of insulin and fMLP receptors with the participation of tyrosine phosphorylation [103]. The identified phosphorylated proteins in insulin pathway during our analysis are Cbl, Rps6, and Inpp5d. The Cbl protein, regulated in iIR compared to Ctrl, has been known for its role in apoptosis, focal adhesion, and that its inhibition protects the heart against IR injury and minimizes the ubiquitination and subsequent degradation of EGFR and monocyte death in response to injury [104,105]. 

Another phosphorylated protein of the pathway, Ribosomal protein S6, regulated in iIR compared to control, Lap, and iIPC, was found common in both the insulin and mTOR pathways during our analysis. The phosphorylation of RpS6 proteins controls the cell size and also regulates the glucose homeostasis [106]. It has been found that IPC results in the phosphorylation of RpS6 protein, which interacts with components of mTOR complex 2 (mTORC2), increases mTORC2 activity, and plays a key role in cardioprotection after ischemia reperfusion [107]. The mTOR-signaling pathway plays an important role in various neutrophil functions. It has been reported that the loss of mTORC2 in neutrophils caused by silencing of Rictor results in failure of neutrophil polarization towards the chemoattractants [108]. It plays a key function in the actin cytoskeleton and also controls the chemotaxis and directional movement of the neutrophils [109,110,111].

Chemokine signaling pathway was found with significant enrichment for the four phosphorylated proteins from clusters 3 and 5 with regulated phosphosites after iIR (compared to Ctrl) and iIPC (compared to iIR). One of the assigned proteins, dock2, has been reported for its role in chemokine dependent activation in F-actin polymerization present at the leading edge of the migrating cell [112]. It also affects other cellular activities of the cell, like chemotaxis, production of superoxide, and formation of extracellular traps [113]. Paxillin, another protein of the chemokine-regulated pathway, was identified in this study with regulated phosphorylation at tyrosine and has been previously documented for its role to be involved in adhesion due to a domain interacting with the C-terminus of focal adhesion kinase (FAK) [114]. Activation of neutrophils due to tyrosine phosphorylation of Fgr domain has already been discussed, causing respiratory burst and polymerization of F-actin [115]. The rolling and recruitment of leukocytes depends on E-selectin, and it has been found to be slow and was associated with rasgrp2 [116]. Our analysis confirms the key role of phosphorylation events in dock2, Fgr, paxillin, and rasgrp2 in neutrophil activation and highlights the need of a keen understanding of the involved processes. Different pro-inflammatory mediators and cytokines are released as a result of Fc receptor interaction with Ig ligands that triggers a process of phagocytosis called antibody-dependent cell-mediated cytotoxicity [117] (ADCC). Another interesting pathway that was found significantly enriched for the phosphorylated proteins from cluster 3 and 5 is Fc-γ R-mediated phagocytosis. The enriched proteins from the two clusters, 3 and 5, to the Fc-γ R-mediated phagocytosis are dock2, Inpp5d, and marcks. Dock2 has already been known for its key role in F-actin architecture and motility and polarity during chemotaxis in neutrophil [118,119]. Marcks (myristoylated alanine-rich C kinase substrate) is a PKC target and has a role in cross-linking actin filaments which is involved in macrophage phagocytosis. Marcks deficiency resulted in the prevention of such processes [120]. 

Focal adhesion is an important phenomenon for the immune cells, and the proteins that affect this process are also very important in neutrophils. In our pathway analysis, the phosphorylated proteins enriched for the focal adhesion include Zyx and Pxn, regulated in iIR and iIPC vs. Ctrl and Ppp1r12a regulated in iIR and iIPC vs. Ctrl and in iIR vs. iIPC. Studies have shown acute inflammatory response as a result of high protein efflux and adhesion of PMNs in postcapillary venules [121]. Intestinal ischemia and reperfusion in mouse overexpresses P-selectin and E-selectin on neutrophils and ECs, respectively. Reduced neutrophil rolling and adhesion as result of P-selectin blockage attenuates the injury [122]. Focal adhesion-like structures have been observed in the primed neutrophils of severely injured patients that regulate various neutrophil functions [123]. A large number of adhesion associated components including various proteins are linked to actin like vinculin, alpha-actinin, talin, and zyxin among others [124]. The regulation of the phosphorylated proteins that are related to adhesion and migration reflects modulation of the major events within neutrophils after iIR and iIPC. 

The SNARE protein vesicle-associated membrane protein 8 (Vamp8 - Q9WUF4) was identified in cluster 3 in its phosphorylated form. Phosphorylation has already been described as a suppressor of its function in mast cells [125]. SNARE proteins play a major role on extracellular release of neutrophil granular contents [126], which is critical for innate immune responses and the outburst of inflammation but can also cause huge organ injury during chronic or acute inflammatory conditions like that triggered by iIR [127]. Vamp8, also known as endobrevin was initially characterized as involved in fusion between early and late endosomes and can interact with syntaxin-7 (also identified as phosphorylated in cluster 3), syntaxin-8, and Vtib to form an endosomal fusion complex [128]. More recently, Vamp8 was demonstrated to be involved in regulated exocytosis by different cell types, including granulocytes, and to be localized to azurophilic granules and in low-density membrane-enriched fractions [129]. Vamp8 phosphorylation allows vesicle docking but suppresses secretion by preventing fusion with the plasma membrane [125]. Interestingly, analysis of Vamp8-deficient mice revealed a major physiological role for this protein in inflammatory conditions such as pancreatitis, as the Vamp8^-/-^ mice showed a significantly attenuated inflammatory response compared to wild-type animals [130]. It is interesting to note that in the present study, the down-regulation of phosphorylated Vamp8 by iIR was prevented by iIPC, thus phosphorylated Vamp8 may be an important target to investigate IPC protection against IR injury. Furthermore, we found phosphorylated forms of Rab7a, Rab1a, Rab27a, and Rab31 up-regulated in iIR compared to other groups (cluster 5). The interaction between Rab proteins and SNAREs plays a pivotal role in neutrophil exocytosis [131]. As iIR seems to increase phosphorylation of Rabs, one could speculate a regulatory mechanism of Vamp8 and Rabs by phosphorylation that connects them to iIR injury and iIPC protection.

## 4. Materials and Methods 

### 4.1. Chemicals and Reagents

Inhibitor Mix and Quant-iT™ Protein Assay Kit were from GE Healthcare Biosciences (Pittsburgh, USA). TEAB was from Sigma Aldrich, Switzerland. Protease PhosSTOP from Roche, Germany. Iodoacetamide was from GE Healthcare (Amersham, UK). Trypsin was from Promega (Madison, WI). DTT and SDS were from GE Healthcare Bio-Sciences (Uppsala, Sweden). iTRAQ reagents from Applied Biosystems (Foster City, CA). Vivacon 30KDa filters were from Sartorius Stedim Biotech (Goettingen, Germany). TSKGel Amide-80 2 mm, 3 mm particle size was from Tosoh Bioscience (Stuttgart, Germany). Empore C8 extraction disk was purchased from 3 M Bioanalytical Technologies (St. Paul, MN). TSKGel Amide-80 2 mm, 3 mm particle size was from Tosoh Bioscience (Stuttgart, Germany). Poros Oligo R2 and R3 reversed-phase materials from PerSeptive Biosystems (Framingham, MA). ReproSil-Pur C18 AQ 3 µm material was from Dr. Maisch (Ammerbuch-Entringen, Germany). Low-binding polypropylene microtubes were from Sorenson Bioscience (Salt Lake City, UT). TFA was from Fluka (St. Louis, MO). All other solvents and chemicals used in this work were HPLC grade or higher. 

### 4.2. Preparation of Experimental Subjects and Sample Collection

A total of 40 male Wistar rats (weighing 250–350 g) were collected from the animal house facility, Faculty of Medicine, University of São Paulo, Brazil. The rats were randomly allocated into the four groups, i.e., control group (Ctrl, without any surgical procedure), sham laparotomy group (Lap, without clamping of any artery but receiving the same surgical procedure), intestinal ischemia and reperfusion group (iIR, subjected to 120 min of reperfusion preceded by 45 min of superior mesenteric artery occlusion (SMAO)), and intestinal ischemic preconditioning group (iIPC, subjected to 10 min of SMAO followed by 10 min of reperfusion immediately prior to 45 min of ischemia and 120 min of reperfusion as in iIR). The project was approved by the ethical committee of FMUSP (Protocol No. 8186), all the surgical procedures and sample preparation were performed in the Laboratory of Surgical Physiopathology (LIM-62), Department of Surgery, FMUSP. To ensure a high sample quality and absence of pre-existing inflammatory responses, hemocytometric analysis using granulocytes count as selection criterion was performed before the surgical procedures and sample collection as previously reported [132]. 

### 4.3. Neutrophil Isolation, Protein Digestion and iTRAQ Labelling

About 10–12 mL of blood was collected from the heart and processed for the neutrophil isolation by using Ficoll density gradient method as reported [31]. After the neutrophil isolation and counting, 3 × 10^6^ neutrophils were tip sonicated for protein extraction in a 200 μL of lysis buffer (2% SDS, 20mM TEAB, 10 mM DTT and protease inhibitor Mix). After sonication, the protein samples were heated at 80 °C for 10 min in water bath and centrifuged, and the protein concentration was determined by using Quant-iT Protein Assay Kit (Thermo Scientific, Waltham, MA, USA). The samples were combined in pools of two to get the required protein concentration, therefore, from ten rats we got five biological replicates. To remove SDS from the protein samples, Vivacon spin filters (Sartorius Stedim, CA, USA) of 30 KDa were used according to Wisniewski et al. [133]. The DTT reduced samples were alkylated with 20 mM of Iodoacetamide in the dark, in 400 μL of 1% SDC solution at room temperature. The samples were washed with 400 μL of 1% SDC solution followed by on filter trypsin digestion in 1:50 (trypsin:protein) ratio at 37 °C overnight. After digestion, the samples were acidified with TFA to a final concentration of 0.1% and centrifuged at 15,000× *g* for 10 min. The peptides were purified using homemade microcolumns packed with Poros Oligo R2/R3 resins (~1 cm long) in p200 tips and the concentration was determined by amino acid analysis [134,135]. After quantification, 100 μg of purified peptides was picked into new low binding tubes from all the four conditions and replicates and dried down. The peptides were reconstituted in 20 μL of iTRAQ dissolution buffer in accordance with the manufacturer protocol. Peptides from Ctrl were labeled with 114, Lap with 115, iIR with 116, and iIPC with 117. Labeling was confirmed by MALDI and samples were multiplexed in 1:1:1:1 ratio.

### 4.4. Enrichment of Phosphopeptides (TiO_2_-SIMAC-HILIC Procedure)

Enrichment of the phosphopeptides from the iTRAQ labeled samples was performed using the following steps:1:First TiO_2_ purification2:IMAC purification3:Second TiO_2_ purification

#### 4.4.1. First TiO_2_ Purification

The enrichment of phosphorylated peptides was performed using the titanium dioxide chromatography as previously reported [136,137,138]. Briefly, a 400 μg of the labeled peptides was reconstituted in 800 μL of loading buffer (5% TFA (*v*/*v*), 1M glycolic acid, and 80% acetonitrile (*v*/*v*) (ACN)) in low binding tubes. The samples were incubated with 0.6 mg of the TiO_2_ beads per 100 μg of peptides with constant high agitation at room temperature for 15 min. The samples were centrifuged (6000 rpm for 30 sec), and the supernatant was incubated with half of the amount of TiO_2_ used in the first incubation in a new tube. The process was repeated to recover the maximum of the phosphopeptides possible. The flow-through (FT) from the second TiO_2_ incubation was transferred to a new low binding tube for further downstream analysis of the total proteome. The TiO_2_ beads from two incubations were pooled together by using 100 μL of the loading buffer. The beads were washed with 100 μL of washing buffer-1 (1% TFA, 80% ACN) and 100 μL washing buffer-2 (0.2% TFA and 10% ACN) and dried down for 10–15 min. The bound peptides from the TiO_2_ surface were eluted with 100 μL of the elution buffer (60 μL ammonia solution (28%) in 940 μL Milli-Q, pH 11.3) with high agitation for 15 min at room temperature. To maximize the phosphopeptides elution and recovery, the elution step was repeated for the second time. The peptides solution was passed through a C8 stage tip to remove the TiO_2_ beads and the samples were lyophilized completely. 

#### 4.4.2. Enzymatic Deglycosylation

As the TiO_2_ chromatography also enriches the sialylated glycopeptides in parallel to the phosphorylated peptides, in order to get rid of the sialylated glycopeptides, the peptides were processed for the deglycosylation. The lyophilized peptides were re-solubilized in 50 μL of 20mM of TEAB and 2 μL of 1U/μL PNGase F, and 0.5 μL of 1U Sialidase A was added to the samples. The enzymatic reaction was performed at 37 °C over night in a wet chamber [139].

#### 4.4.3. IMAC Purification of Multi-Phosphorylated Peptides

The identification of the multi-phosphorylated peptides is usually compromised compared to the monophosphorylated peptides due to their low abundance. To increase identification of the multi-phosphorylated peptides, IMAC purification was performed for the enrichment. The peptides solution from the deglycosylation was acidified by adding 1 μL of 10% TFA and diluted with 200 μL of 50% ACN/0.1% TFA, and the pH was adjusted to 1.6–1.8. Then 200 μL of IMAC beads were picked and washed twice with 50% ACN/0.1% TFA. The washed beads were added to the peptide solution and incubated for 30 min at room temperature. The samples were centrifuged, and half of the supernatant was transferred to a new tube whereas the remaining supernatant with the IMAC beads was shifted to the 200 μL GeLoader tip (constricted at the tip to hold the beads). The remaining supernatant was passed through with help of a syringe and the IMAC beads column was packed. The beads were washed with 100 μL of 50% ACN/0.1% TFA. The beads were washed for the second time with 70 μL of 20% ACN/1% TFA and the FT was collected with the supernatant containing monophosphorylated peptides. This step was performed slowly to allow the monophosphorylated peptides to fall off the IMAC beads and still allow the multiphosphorylated peptides to be retained. The multiphosphorylated peptides were eluted with 100 μL of ammonia (as for the first TiO_2_ elution) directly in a p200 stage tip with Poros Oligo R3 materials and acidified with 2 μL of 10% TFA prior to the sample touching the R3 material of the column. The multiphosphorylated peptides were R3 purified and lyophilized for analysis by mass spectrometry. 

#### 4.4.4. Second TiO_2_ Purification

The SIMAC FT was dried completely and re-suspended in 200 μL of 70% ACN/2% TFA solution. The peptides were incubated twice with the same amount of the TiO_2_ beads as in the first time. The beads were washed with 100 μL washing buffer-1 (80% ACN/1% TFA) and 100 μL of washing buffer-2 (10% ACN/0.2% TFA). All the supernatants/FTs were pooled together containing the sialylated glycopeptides (de-glycosylated). The beads were dried for 10 min, and the monophosphorylated peptides were eluted with ammonia. The peptides were passed through a C8 plug to remove the TiO_2_ spheres. The sample was R3 purified and HILIC fractionated.

#### 4.4.5. HILIC Fractionation of Total and Mono-Phosphorylated Peptides

The HILIC fractionation of the purified total peptides (first TiO_2_ FT) and mono-phosphorylated peptides (second TiO_2_ eluate) was performed as previously documented [140]. Briefly, the dried peptides were re-solubilized in a buffer (90% ACN/0.1% TFA) and 40 μL of the sample was loaded onto an in-house packed TSK gel Amide-80 HILIC 320 μm × 170 mm capillary HPLC column connected to an Agilent 1200 HPLC system. The gradient used for the elution of the peptides was from 90% ACN/0.1% TFA to 60% ACN/0.1% TFA for 35 min at a flow rate of 6 μL/min. The fractions were automatically collected at 1 min interval in a microwell plate after UV detection at 210 nm. The fractions were dried completely and analyzed by the mass spectrometer. 

### 4.5. Nano-Liquid Chromatography Tandem Mass Spectrometry (nLC-MS/MS)

The multi-phosphorylated and fractionated mono-phosphorylated peptides were analyzed by a Proxeon Easy-nLC system (Thermo Fischer Scientific, Odense, Denmark), coupled with LTQ-Orbitrap Velos mass spectrometer (Thermo Fischer Scientific). The samples were loaded onto an 18 cm long home-made reversed phase analytical column (75 μm inner diameter) packed with ReproSil-Pur C18 AQ 3 μm material (Dr. Maisch, Ammerbuch Entringen, Germany). Peptidic fractions eluted from the nLC directly into the ion source of the mass spectrometer, that was operated in data-dependent acquisition mode. The first MS scan (MS1) was carried out in the m/z range of 400–1200, at a resolution of 30,000. From each MS1 scan, the seven most intense components above a signal of 20,000 were submitted to HCD fragmentation at NCE of 36% and analyzed in the Orbitrap detector at a resolution of 7500. The spectra were saved as raw files for further analyses and were deposited at theProteomeXchange [141] consortium via the Mass Spectrometry Interactive Virtual Environment (MassIVE) platform [142] under the ID PXD016182, MSV000084545. 

### 4.6. Database Searching and Bioinformatics Analysis

The raw files resulted from the mass spectrometer were analyzed by Proteome Discoverer version 1.4.0.288 (Thermo Fischer Scientific). The .mgf files were created from the tandem MS/MS spectra, and searches were made against the UniProt rodents’ database using Mascot (v2.3.2, Matrix Science, London, UK). The parameters used for searching of the database were: precursor mass tolerance 10 ppm, fragment (MS/MS) mass tolerance 0.05 Da, two missed cleavages were allowed, trypsin was selected as digestion enzyme, variable modifications included oxidation (M), deamidation, and for phosphopeptides serine, threonine and tyrosine were added to the searches. The carbamidomethylation of cysteine residue was selected as a fixed modification. The results were filtered for 1% false discovery rate (FDR) using Percolator as validator [143]. Further stringent filters were applied for the phosphopeptides by excluding all with phosphoRS 3.0 probability lower than 95%. 

For statistical analysis, an in-house script developed under statistical package R software was used. The iTRAQ intensities from all the five biological replicates were log2 transformed and median normalized. One peptide measurement was allowed by choosing mean instead of median in RRollup function of DanteR package for the multiple measurements of the same peptide [144]. Limma and Rank Products were applied, providing sufficient power to deal with low replicate numbers and additional missing values [145,146]. Both statistical tests were carried out against label 114 (Ctrl) and corrected for multiple testing. The protein and phosphopeptides with q-value below 0.05 (5% FDR) were considered as regulated. 

Cluster analysis was performed, by calculating the mean over all five replicated values for each condition. A single dataset was created after merging the phosphorylated peptides and proteins data. We applied Fuzzy c-mean clustering after determining the value of the fuzzifier and obtaining the number of clusters according to Schwämmle [147]. A standard principle component analysis (PCA) analysis was performed using R to check similarity among the replicates of the same group and variability between different conditions. 

ProteinCenter (Thermo Fischer Scientific, Waltham, USA) was used to interpret the results at protein level (statistical GO *slim* functional categorization with 5% FDR and number of domains). WebGestalt was used with default parameters for KEGG pathways analysis [148]. Using GPS software, we performed a kinase recognition site analysis on protein sequences, and *Rattus norvegicus* was selected as database. The kinase-substrate prediction analysis of the significantly regulated phosphopeptides was performed in iGPS 1.0 [42] against *M. musculus* database using high prediction confidence, and string/experiment was selected to find the interactions. MotifX algorithm was used with windows of 17 amino acids for the analysis of sequence motif enrichment of the phosphorylation sites [149]. Phosphorylation in the regulating domains of kinases and phosphatases were checked manually in the NCBI database (www.nvbi.nlm.nih.gov/cdd).

## 5. Conclusions

Our deep analyses of the rat neutrophil phosphoproteome revealed many phosphorylation events that occur in various domains and motifs regions of interesting kinases and phosphatases. Most of these proteins regulate apoptosis, degranulation, and migration in the neutrophils. Our experimental design validated the role of previously studied kinases and extended the knowledge of neutrophil biology with addition of new possible players identified for the first time in our analysis. Further investigation of these new candidates with their regulating pathways will open new windows to mimic the tissues injury after iIR by the activated neutrophils. Furthermore, changes in neutrophil behavior might be due to phosphorylation events taking place in PKCδ, NDR1/Stk38, or proteins that present a phosphorylation profile compatible with modulation by iIR and prevention of such modulation by iIPC, such as Inpp5d/Ship and Vamp8. Our analysis shows that most of the important predicted kinases that might regulate other kinases identified here in neutrophils from iIR and iIPC belong to the CAMK and CMGC families. The understanding of possible interaction among different kinases and the use of documented inhibitors will help to define possible therapeutic targets to decrease the neutrophil activation and tissue injury in iIR surgical procedures.

## Figures and Tables

**Figure 1 ijms-21-05799-f001:**
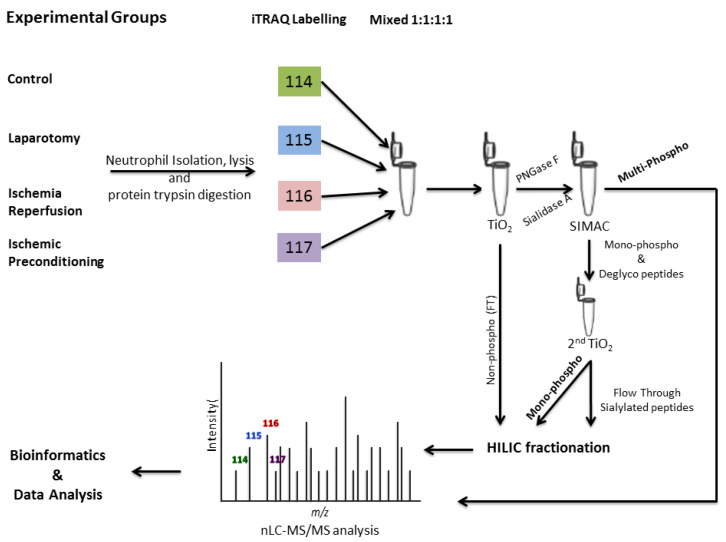
Schematic representation of the experimental workflow. Isolation of neutrophils by Ficoll density gradient from control (Ctrl), laparotomy (Lap), intestinal ischemia/reperfusion (iIR), and intestinal ischemic preconditioning (iIPC) groups followed by protein extraction and digestion. The iTRAQ labeling 114 for Ctrl, 115 for Lap, 116 for iIR, and 117 for iIPC was used. Mono- and multi-phosphopeptides enrichment was performed using TiSH (TiO_2_ SIMAC HILIC fractionation). LTQ Orbitrap mass spectrometry (MS) analysis followed bioinformatics tools.

**Figure 2 ijms-21-05799-f002:**
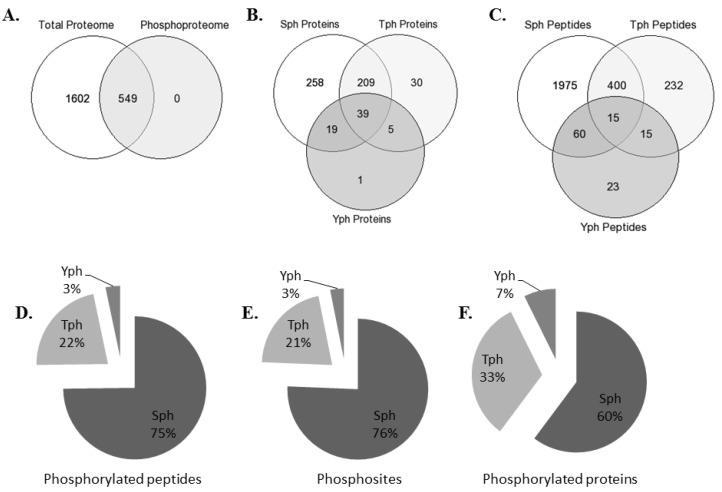
Statistical overview of total and phosphoproteome. (**A**) Overlap between proteome and phosphoproteome. (**B**) Overlap between phosphorylated serine, threonine, and tyrosine containing phosphoproteome. (**C**) Overlap between serine, threonine, and tyrosine phosphorylated peptides. (**D**) Phosphorylated peptides distribution. (**E**) Phosphosites distribution. (**F**) Phosphorylated proteins distribution.

**Figure 3 ijms-21-05799-f003:**
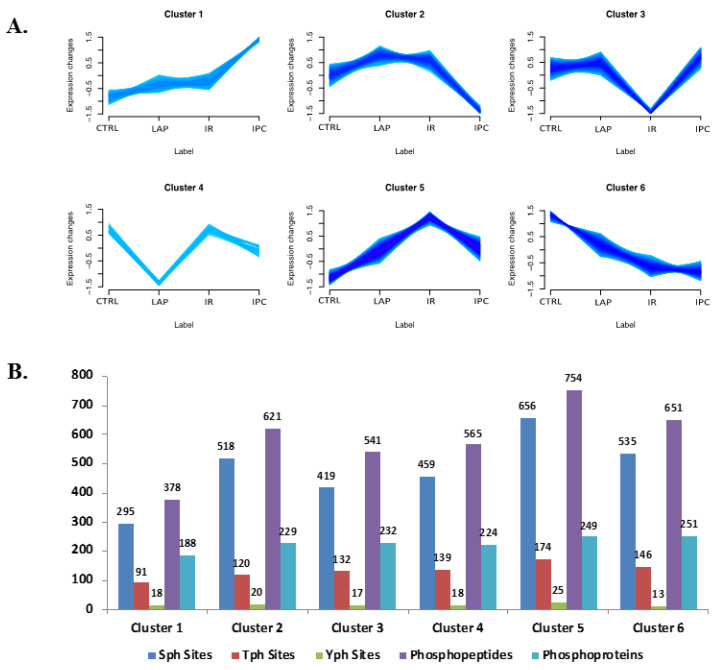
(**A**) Cluster-based relative abundance profiles of the identified proteins and phosphopeptides in Ctrl, Lap, iIR, and iIPC from rat neutrophil. (**B**) Phosphosites, phosphopeptides, and phosphoproteins distribution across the six clusters.

**Figure 4 ijms-21-05799-f004:**
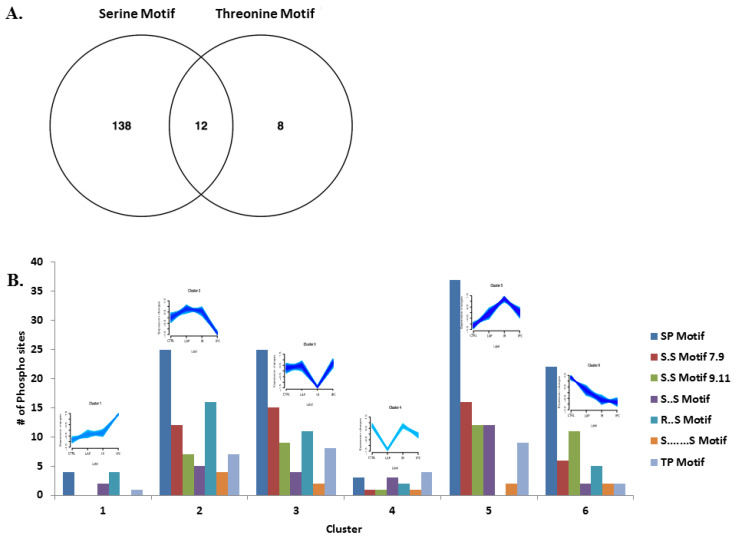
Analysis of phosphorylation motifs grouped by regulatory trends. (**A**) Venn diagram of phosphorylated proteins with enriched serine and threonine motifs. (**B**) Distribution of the enriched serine and threonine motifs abundance across six clusters.

**Figure 5 ijms-21-05799-f005:**
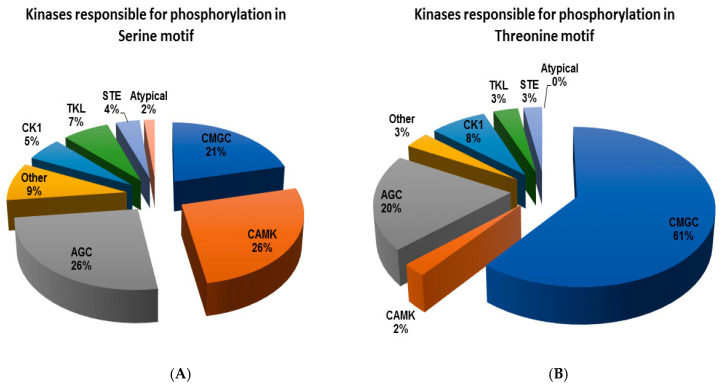
Distribution of the predicted kinase families responsible for the phosphorylation event in the serine (**A**) and threonine (**B**) motifs.

**Table 1 ijms-21-05799-t001:** Kinases and Phosphatases with regulated phosphorylation.

Protein ID	Description	Gene Symbol	Regulated Phosphopeptide in Domain	Phospho Reg. Cluster	Sig. Regulation	Phosphosite in Protein	Enzyme Code	Domain Containing the Phosphopeptide	Domain Description
**Kinases**									
Q6P6U0	Tyrosine-protein kinase Fgr	Fgr	LIVDDEYphNPQQGTKFPIK	2	IR vs. Ctrl	Y400	EC: 2.7.10.2	PTKc_Src_Fyn_like	Catalytic domain of a subset of Src kinase-like Protein Tyrosine Kinases
								Pkinase_Tyr	Protein tyrosine kinase
Q64303	Serine/threonine-protein kinase PAK 2	Pak2	--	--	--	--	EC: 2.7.11.1	--	--
E9PTG8	Serine/threonine-protein kinase 10	Stk10	--	--	--	--	EC: 2.7.11.1	--	--
Q63531	Ribosomal protein S6 kinase alpha-1	Rps6ka1	--	--	--	--	EC: 2.7.11.1	--	--
Q9WUT3	Ribosomal protein S6 kinase alpha-2	Rps6ka2	--	--	--	--	EC: 2.7.11.1	--	--
P26817	Beta-adrenergic receptor kinase 1	Adrbk1	NKPRSphPVVELSK	2	IR vs. Ctrl	S670	EC: 2.7.11.15	PH_GRK2_subgroup	G Protein-Coupled Receptor Kinase 2 subgroup pleckstrin homology (PH) domain
									G-beta gamma binding site
Q63433	Serine/threonine-protein kinase N1	Pkn1	--	--	--	--	EC: 2.7.11.13	--	--
P09215	Protein kinase C delta type	Prkcd	SPSDYSNFDPEFLNEKPQLSphFSDK	6	IR vs. Ctrl	S643	EC: 2.7.11.13	STKc_nPKC_delta	Catalytic domain of the Serine/Threonine Kinase, Novel Protein Kinase C delta
P09215	Protein kinase C delta type	Prkcd	SPSDYSNFDPEFLNEKPQLSFSphDK	6	IR vs. Ctrl	S645	EC: 2.7.11.13	STKc_nPKC_delta	Catalytic domain of the Serine/Threonine Kinase, Novel Protein Kinase C delta
Q64725	Tyrosine-protein kinase SYK	Syk	--	--	--	--	EC: 2.7.10.2	--	--
O08815	STE20-like serine/threonine-protein kinase	Slk	--	--	--	--	EC: 2.7.11.1	--	--
Q6P9R2	Serine/threonine-protein kinase OSR1	Oxsr1	--	--	--	--	EC: 2.7.11.1	--	--
Q91VJ4	Serine/threonine-protein kinase 38	Stk38	FEGLTphAR	1	IR vs. IPC	T452	EC: 2.7.11.1	STKc_NDR1	Catalytic domain of the Serine/Threonine Kinase, Nuclear Dbf2-Related kinase 1
**Phosphatases**									
P81718	Tyrosine-protein phosphatase non-receptor type 6	Ptpn6	DLSphGPDAETLLK	2	IR vs. Ctrl	S12	EC: 3.1.3.48	SH2_N-SH2_SHP_like	N-terminal Src homology 2 (N-SH2) domain found in SH2 domain Phosphatases (SHP) proteins
P97573	Phosphatidylinositol 3,4,5-trisphosphate 5-phosphatase 1	Inpp5d	TGIANTphLGNK	3	IR vs. IPC	T519	EC: 3.1.3.86	INPP5D_SHIP1-INPP5D	Catalytic inositol polyphosphate 5-phosphatase (INPP5d) domain of SH2 domain
									Putative catalytic site
									Putative active site
									Putative Mg binding site
									Putative PI/IP binding site
B2GV87	Receptor-type tyrosine-protein phosphatase epsilon precursor	Ptpre	--	--	--	--	EC: 3.1.3.48	--	--
P04157	Receptor-type tyrosine-protein phosphatase C isoform 4 precursor	Ptprc	--	--	--	--	EC: 3.1.3.48	--	--

**Table 2 ijms-21-05799-t002:** Enriched motifs found in the regulated phosphoproteins.

Class/ID	Significantly Regulated Phosphopeptide	Significant Regulation	Modified pos. in Peptide	Data Source 1	Position in Protein	Cluster	Motif
**Kinases**							
Q6P6U0	EDVGLEGDFRSphQGAEER	IR vs. Ctrl	S11	RAT	S25	5	
	LIVDDEYphNPQQGTKFPIK	IR vs. Ctrl	Y7	RAT	Y400	2	
	SphSSISPQPISPAFLNVGNIR	IR vs. Ctrl	S1	RAT	S41	2	S.S
	SphSSISPQPISphPAFLNVGNIR	IR vs. Ctrl	S1	RAT	S41	5	S.S
	SSphSISPQPISphPAFLNVGNIR	IR vs. Ctrl	S2	RAT	S42	5	S.......S
	SSSISphPQPISphPAFLNVGNIR	IR vs. Ctrl	S5	RAT	S45	5	SP
	SSSphISphPQPISPAFLNVGNIR	IR vs. Ctrl	S3	RAT	S43	5	S.S
	SSSphISPQPISphPAFLNVGNIR	IR vs. Ctrl	S3	RAT	S43	5	S.S
	EDVGLEGDFRSphQGAEER	IR vs. Lap	S11	RAT	S25	5	
Q64303	FYDSphNTVK	IR vs. Ctrl	S4	RAT	S132	2	
	YLSphFTPPEK	IR vs. Lap	S3	RAT	S141	6	
E9PTG8	ILRLSphTFEK	IR vs. Ctrl	S5	RAT	S13	6	
	LSphTFEK	IR vs. Ctrl	S2	RAT	S13	6	
	LSphTFEK	IR vs. Lap	S2	RAT	S13	6	
	LSTphFEK	IR vs. IPC	T3	RAT	T14	1	
Q63531	KLPSphTTL	IR vs. Ctrl	S4	RAT	S732	2	
Q9WUT3	LEPVLSphSSLAQR	IR vs. Ctrl	S6	MOUSE	S716	3	S.S
	LEPVLSphSSLAQR	IR vs. IPC	S6	MOUSE	S716	3	S.S
P26817	NKPRSphPVVELSK	IR vs. Ctrl	S5	RAT	S670	2	SP
Q63433	SGSphLSGR	IR vs. Ctrl	S3	RAT	S377	2	S.S
P09215	SPSDYSNFDPEFLNEKPQLSFSphDK	IR vs. Ctrl	S22	RAT	S645	6	S.S
	SPSDYSNFDPEFLNEKPQLSphFSDK	IR vs. Ctrl	S20	RAT	S643	6	S.S
	SPSDYSNFDPEFLNEKPQLSphFSDK	IR vs. Lap	S20	RAT	S643	6	S.S
	SPSDYSNFDPEFLNEKPQLSphFSDK	IR vs. IPC	S20	RAT	S643	6	S.S
Q64725	SYSphFPKPGHK	IR vs. Ctrl	S3	RAT	S291	6	S.S
O08815	TKDSGSphVSLQETR	IR vs. Lap	S6	RAT	S778	3	S.S
Q6P9R2	AAISQLRSphPR	IR vs. IPC	S8	MOUSE	S359	5	SP
	RVPGSphSphGRLHK	IR vs. IPC	S5	MOUSE	S324	3	
Q91VJ4	FEGLTphAR	IR vs. IPC	T5	MOUSE	T452	1	
**Phosphatases**							
P81718	DLSphGPDAETLLK	IR vs. Ctrl	S3	RAT	S12	2	R..S
	TphSSKHKEEVYENVHSK	IR vs. Ctrl	T1	RAT	T557	5	
	DLSphGPDAETLLK	IR vs. Lap	S3	RAT	S12	2	R..S
P97573	DSSLGPGRGEGPPTphPPSQPPLSPK	IR vs. Ctrl	T14	RAT	T963	2	TP
	GEGPPTphPPSQPPLSphPK	IR vs. Ctrl	T6	RAT	T963	5	TP
	GEGPPTphPPSQPPLSphPKK	IR vs. Ctrl	T6	RAT	T963	5	TP
	KEQESphPK	IR vs. Ctrl	S5	RAT	S1037	2	SP
	KEQESphPK	IR vs. Lap	S5	RAT	S1037	2	SP
	GEGPPTphPPSQPPLSPK	IR vs. IPC	T6	RAT	T963	2	TP
	TGIANTphLGNK	IR vs. IPC	T6	RAT	T519	3	
B2GV87	SPSphGPKK	IR vs. Ctrl	S3	RAT	S106	3	S.S
	SPSphGPKK	IR vs. Lap	S3	RAT	S106	3	S.S
P04157	ANSphQDKIEFHNEVDGAK	IR vs. Lap	S3	RAT	S1209	6	
	KANSphQDK	IR vs. Lap	S4	RAT	S1209	6	
	KANSphQDKIEFHNEVDGAK	IR vs. Lap	S4	RAT	S1209	6	
	ANSphQDKIEFHNEVDGAK	IR vs. IPC	S3	RAT	S1209	6	
	KANSphQDK	IR vs. IPC	S4	RAT	S1209	6	
	KANSphQDKIEFHNEVDGAK	IR vs. IPC	S4	RAT	S1209	6	

**Table 3 ijms-21-05799-t003:** Enriched motifs found in the regulated phosphoproteins.

Acc. No.	Phosphosite	Predicted Kinase	Phosphorylated Peptide	Score	Cutoff	Motif	Cluster
Kinases							
Q6P6U0	Y400	TK/Src	LIVDDEYphNPQQGTKFPIK	24.645	1.63	--	6
	Y400	TK/Tec*	LIVDDEYphNPQQGTKFPIK	23.341	3.584	--	
	Y400	TK/Jak	LIVDDEYphNPQQGTKFPIK	18.242	8.154	--	
	Y400	TK/FAK*	LIVDDEYphNPQQGTKFPIK	14.808	5.968	--	
	Y400	TK/DDR*	LIVDDEYphNPQQGTKFPIK	11	2.683	--	
	Y400	TK/Syk	LIVDDEYphNPQQGTKFPIK	8.825	2.436	--	
	Y400	TK/Csk*	LIVDDEYphNPQQGTKFPIK	6.778	4.886	--	
	Y400	TK/Abl*	LIVDDEYphNPQQGTKFPIK	5.422	4.747	--	
	Y400	TK/Met*	LIVDDEYphNPQQGTKFPIK	4.607	2.379	--	
	Y400	TK/VEGFR*	LIVDDEYphNPQQGTKFPIK	4.556	3.654	--	
	Y400	TK/Alk	LIVDDEYphNPQQGTKFPIK	3.667	3.333	--	
	Y400	TK/PDGFR*	LIVDDEYphNPQQGTKFPIK	3.646	2.352	--	
	Y400	TK*	LIVDDEYphNPQQGTKFPIK	2.516	0.166	--	
P26817	S670	CMGC/MAPK	NKPRSphPVVELSK	42.735	35.046	SP	4
	S670	AGC/PDK1	NKPRSphPVVELSK	5.222	2.257	SP	
	S670	CMGC*	NKPRSphPVVELSK	1.356	0.963	SP	
	S670	CMGC/DYRK*	NKPRSphPVVELSK	1.333	1.276	SP	
	S670	AGC/PKC	NKPRSphPVVELSK	0.384	0.236	SP	
P09215	S643	CAMK/RAD53*	SPSDYSNFDPEFLNEKPQLSphFSDK	13.075	7.385	S.S	5
	S643	AGC/PKC	SPSDYSNFDPEFLNEKPQLSphFSDK	0.905	0.236	S.S	
	S643	Atypical/PDHK*	SPSDYSNFDPEFLNEKPQLSphFSDK	4.405	3.075	S.S	
	S645	CAMK/CAMK1*	SPSDYSNFDPEFLNEKPQLSFSphDK	3.259	2.488	S.S	
	S645	TKL/STKR*	SPSDYSNFDPEFLNEKPQLSFSphDK	2.938	2.562	S.S	
Q91VJ4	T452	TKL*	FEGLTphAR	4.648	4.354	--	4
	T452	AGC/PKC	FEGLTphAR	0.608	0.236	--	
Phosphatases							
P81718	S12	CAMK/PHK*	DLSphGPDAETLLK	22.269	9.527	R..S	4
	S12	CMGC/CK2*	DLSphGPDAETLLK	12.467	9.894	R..S	
	S12	CMGC/CLK*	DLSphGPDAETLLK	5.375	4.3	R..S	
P97573	T519	Other/TLK*	TGIANTphLGNK	6.25	5.775	--	5
	T519	Other/TTK*	TGIANTphLGNK	5.188	5.009	--	

Predicted Kinases with * shows the absence of inhibitors in the database.

**Table 4 ijms-21-05799-t004:** Overrepresented KEGG pathways associated to regulated phosphoproteins.

KEGG Pathway	Accession No.	Gene Symbol	Description	No. of Phosphopeptides
**Spliceosome** (*C = 135; O = 4; E = 0.26; R = 15.49; rawP = 0.0001; adjP = 0.0017*)				
	P26369	U2af2	U2 small nuclear ribonucleoprotein auxiliary factor (U2AF) 2	1
	Q62093	Srsf2	serine/arginine-rich splicing factor 2	2
	Q99NB9	Sf3b1	splicing factor 3b, subunit 1	2
	Q3TIX9	Usp39	ubiquitin specific peptidase 39	1
**Chemokine signaling pathway** *(C = 178; O = 4; E = 0.34; R = 11.75; rawP = 0.0004; adjP = 0.0034)*				
	Q8C3J5	Dock2	dedicator of cytokinesis 2	4
	Q66H76	Pxn	paxillin	2
	Q6P6U0	Fgr	Gardner–Rasheed feline sarcoma viral (v-fgr) oncogene homolog	6
	P0C643	Rasgrp2	RAS guanyl releasing protein 2 (calcium and DAG-regulated)	1
**Fc gamma R-mediated phagocytosis** *(C = 91; O = 3; E = 0.17; R = 17.24; rawP = 0.0007; adjP = 0.0040)*				
	Q8C3J5	Dock2	dedicator of cytokinesis 2	4
	P97573	Inpp5d	inositol polyphosphate-5-phosphatase D	2
	P30009	Marcks	myristoylated alanine rich protein kinase C substrate	8
**Insulin signaling pathway** *(C = 131; O = 3; E = 0.25; R = 11.97; rawP = 0.0021; adjP = 0.0089)*				
	P22682	Cbl	Cbl proto-oncogene, E3 ubiquitin protein ligase	2
	P62754	Rps6	ribosomal protein S6	1
	P97573	Inpp5d	inositol polyphosphate-5-phosphatase D	2
**Focal adhesion** *(C = 186; O = 3; E = 0.36; R = 8.43; rawP = 0.0055; adjP = 0.0134)*				
	Q62523	Zyx	zyxin	1
	Q10728	Ppp1r12a	protein phosphatase 1, regulatory subunit 12A	4
	Q66H76	Pxn	paxillin	2
**Phagosome** *(C = 185; O = 3; E = 0.35; R = 8.48; rawP = 0.0055; adjP = 0.0134)*				
	O70257	Stx7	syntaxin 7	2
	P35278	Rab5c	RAB5C, member RAS oncogene family	1
	Q91ZN1	Coro1a	coronin, actin binding protein 1A	1
**mTOR signaling pathway** *(C = 51; O = 2; E = 0.10; R = 20.50; rawP = 0.0043; adjP = 0.0134)*				
	Q9WUT3	Rps6ka2	ribosomal protein S6 kinase polypeptide 2	1
	P62754	Rps6	ribosomal protein S6	1
**Endocytosis** *(C = 230; O = 3; E = 0.44; R = 6.82; rawP = 0.0099; adjP = 0.0153)*				
	Q5FVC7	Acap2	ArfGAP with coiled coil, ankyrin repeat and PH domains 2	1
	P22682	Cbl	Cbl proto-oncogene, E3 ubiquitin protein ligase	2
	P35278	Rab5c	RAB5C, member RAS oncogene family	1
**B cell receptor signaling pathway** *(C = 75; O = 2; E = 0.14; R = 13.94; rawP = 0.0092; adjP = 0.0153)*				
	P97573	Inpp5d	inositol polyphosphate-5-phosphatase D	2
	P81718	Ptpn6	protein tyrosine phosphatase, non-receptor type 6	3
**Long-term potentiation** *(C = 69; O = 2; E = 0.13; R = 15.16; rawP = 0.0078; adjP = 0.0153)*				
	Q10728	Ppp1r12a	protein phosphatase 1, regulatory subunit 12A	4
	Q9WUT3	Rps6ka2	ribosomal protein S6 kinase polypeptide 2	1

C: the number of reference genes in the category, O: the number of genes in the gene set and also in the category, E: the expected number in the category, R: ratio of enrichment, rawP: *p*-value from hypergeometric test, adjP: *p*-value adjusted for multiple tests.

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
