# Peer review of "Phosphoproteomic Analysis of Rat Neutrophils Shows the Effect of Intestinal Ischemia/Reperfusion and Preconditioning on Kinases and Phosphatases"

_ijms, 2020, doi:10.3390/ijms21165799_

Round 1
Reviewer 1 Report
Although the authors have tried to provide some rationale and improved their data presentation, this is not convincing enough. The current manuscript is still very much descriptive and lacks any sort of validation. I agree with the fact that this is more of a hypothesis generating study and that is why is more suitable to a specialized proteomics journal.
Author Response
The authors are thankful to the reviewers and editors for dedicating their time to another round of revision. We confirm that this article generates hypotheses, such as the possible modulation of the effect of IRI by interfering in the phosphorylation state of enzymes such as OSR1, Stk38, Grk2, Pak2 and Ptprc, affecting important neutrophil functions like ROS production, apoptosis, adhesion, actin filament/directional migration, phagocytosis and bacterial killing. Even though we did not perform any inhibition assays, that does not render our results invalid, since we presented statistically significant results, in accordance with many studies in the field [1-4]. In addition, a number of other high quality studies with similar characteristics to ours (i.e. rather descriptive and hypothesis-generating omics studies) have been recently published in IJMS [5-9].
Moreover we believe this article is well suited for the special issue "The Impact of Hypoxia on Neutrophil Signaling and Function" of IJMS, as the article describes and discusses many proteins that have significantly regulated phosphorylation by ischemia/reperfusion and proteins that have such modulation prevented by ischemic preconditioning.
REFS: 1- Schwämmle, V.; León, I.R.; Jensen, O.N. Assessment and Improvement of Statistical Tools for Comparative Proteomics Analysis of Sparse Data Sets with Few Experimental Replicates. J. Proteome Res. 2013, 12, 3874–3883.
2- Schwämmle, V.; Jensen, O.N. A simple and fast method to determine the parameters for fuzzy c–means cluster analysis. Bioinformatics 2010, 26, 2841–2848.
3- Taverner, T.; Karpievitch, Y. V; Polpitiya, A.D.; Brown, J.N.; Dabney, A.R.; Anderson, G.A.; Smith, R.D. DanteR: an extensible R-based tool for quantitative analysis of -omics data. Bioinformatics 2012, 28, 2404–2406.
4- Breitling, R.; Armengaud, P.; Amtmann, A.; Herzyk, P. Rank products: a simple, yet powerful, new method to detect differentially regulated genes in replicated microarray experiments. FEBS Lett. 2004, 573, 83–92. 5- Insight in Adhesion Protein Sialylation and Microgravity Dependent Cell Adhesion—An Omics Network Approach Int. J. Mol. Sci. 2020, 21(5), 1749; https://doi.org/10.3390/ijms21051749 6- Co-Expression Network Analysis of Spleen Transcriptome in Rock Bream (Oplegnathus fasciatus) Naturally Infected with Rock Bream Iridovirus (RBIV) Int. J. Mol. Sci. 2020, 21(5), 1707; https://doi.org/10.3390/ijms21051707
7- Spatial and Pregnancy-Related Changes in the Protein, Amino Acid, and Carbohydrate Composition of Bovine Oviduct Fluid. Int J Mol Sci. 2020 Feb 29;21(5). pii: E1681. doi: 10.3390/ijms21051681.
8- In-Depth Understanding of Camellia oleifera Self-Incompatibility by Comparative Transcriptome, Proteome and Metabolome Int. J. Mol. Sci. 2020, 21(5), 1600; https://doi.org/10.3390/ijms21051600 9- Integrative Transcriptomic and Proteomic Analyses of Molecular Mechanism Responding to Salt Stress during Seed Germination in Hulless Barley Int. J. Mol. Sci. 2020, 21(1), 359; https://doi.org/10.3390/ijms21010359
Reviewer 2 Report
Review of the manuscript
The objective of this current work is to analyze the phosphoproteome of rat neutrophils in the context of intestinal ischemia and reperfusion, and the effect of ischemic preconditioning. The methodology involves induction of ischemia and reperfusion, in the presence or absence of preconditioning. Sham laparotomy and control groups are also involved. Neutrophils were isolated from blood and subjected to proteomic analysis. Analysis yielded several crucial phosphopeptides, motifs of interest, and kinases and phosphatases of interest. Several key targets are identified. The current study is hypothesis generating in its scope, and the following questions are raised:
The neutrophils are isolated from cardiac blood, followed by subsequent processing. Is this representative of the neutrophil population infiltrating the intestines after 120 minutes of reperfusion, as the experimental design suggests? It is possible that the extravasating neutrophils exposed to local inflammatory milieu undergo different phenotypic modulations, and could potentially affect the phosphoproteomic data. The comparisons among groups for phosphoproteins and kinases/phosphatases are not clear. For instance, for Grk2, the discussed differences are between the iIR and Ctrl group. For Stk38, the comparison is between iIR and iIPC. It would be helpful to compare the general implications of these changes between control and lap groups – did the impact of surgery affect the phosphoproteome? Overall, the discussion seems quite fragmented and difficult to follow. A more systematic explanation and organization is warranted. Several interesting pathways were outlined and discussed. What is the significance of insulin signaling, mTOR signaling pathway and Long-term potentiation (which is primarily a neuronal phenomenon) in the neutrophilic response post intestinal ischemia reperfusion? The discussion should further discuss the impact the influence of other cell types (intestinal epithelial cells, crypt cells and other immune cells infiltrating the intestinal mucosa) in the context of iIR, and relate it to the phosphoproteomic data obtained in this study Line 48: Should say depletion of neutrophils Line 50: Should say proteolytic enzyme (not enzymes) secretion
Author Response
Reply to the comments from reviewer 2:
The neutrophils are isolated from cardiac blood, followed by subsequent processing. Is this representative of the neutrophil population infiltrating the intestines after 120 minutes of reperfusion, as the experimental design suggests? It is possible that the extravasating neutrophils exposed to local inflammatory milieu undergo different phenotypic modulations, and could potentially affect the phosphoproteomic data. The comparisons among groups for phosphoproteins and kinases/phosphatases are not clear. For instance, for Grk2, the discussed differences are between the iIR and Ctrl group. For Stk38, the comparison is between iIR and iIPC. It would be helpful to compare the general implications of these changes between control and lap groups – did the impact of surgery affect the phosphoproteome? Overall, the discussion seems quite fragmented and difficult to follow. A more systematic explanation and organization is warranted. Several interesting pathways were outlined and discussed. What is the significance of insulin signaling, mTOR signaling pathway and Long-term potentiation (which is primarily a neuronal phenomenon) in the neutrophilic response post intestinal ischemia reperfusion? The discussion should further discuss the impact the influence of other cell types (intestinal epithelial cells, crypt cells and other immune cells infiltrating the intestinal mucosa) in the context of iIR, and relate it to the phosphoproteomic data obtained in this study Line 48: Should say depletion of neutrophils Line 50: Should say proteolytic enzyme (not enzymes) secretion
As the aim of our study was to analyze the role of neutrophils in the systemic inflammatory response, we did not try to collect just the cells emerging from the ischemic area, but rather cells that were in the circulating pool, including those coming from the mesenteric, pulmonary and hepatic territories. After a reperfusion time of 120 min, these cells had enough time to regulate pathways and crosstalk to other cells and even other tissues, as previously mentioned (PMID 28983053 and PMID 27558331). The interplay between neutrophils and other cell types was also added to the manuscript (Page 2, Lines 50-53). It is likely that neutrophil subpopulations are affected, but that was not within the scope of the present study, and inserting that factor in the experimental design would have multiplied the number of conditions by the number of subpopulations chosen for analysis. That is, indeed, a quite interesting topic for a follow-up study.
Regarding the structure of the discussion, we apologize to the reviewer about the fragmentation. The text was revised and re-organized. Moreover, a discussion about the possible importance of the protein phosphorylation in insulin and mTOR signaling, was included (Page 20, Lines 437-457). Regarding long-term potentiation in the context of the systemic inflammatory response, we did not add information because our pathway analysis was unsupervised, therefore you get all the possible pathways. Although it is very clear that most of the identified pathways are very relevant to the neutrophils biology, at the moment we do not have experimental evidence about LTP for neutrophils.
The authors agree with the reviewer that the surgical trauma affects the neutrophil proteome as well as the phosphoproteome and that was added to the discussion (Page 17, Lines 264-274).
The specific points at lines 48 (currently line 54) and 50 (currently line 55) were modified according to the request of the reviewer.
This manuscript is a resubmission of an earlier submission. The following is a list of the peer review reports and author responses from that submission.
Round 1
Reviewer 1 Report
In this manuscript, Tahir et al., performed a proteomic analysis of rat neutrophils undergoing Intestinal ischemia reperfusion injury (iIRI) or ischemic preconditioning (iIPC) or sham laparotomy group by multi-step enrichment of phosphopeptides, isobaric labeling and LC-MS/MS analysis. They further used bioinformatics analysis to determine phosphosite and phosphopeptide abundance and clustering, as well as kinases and phosphatases sites and domains. In this analysis, the author found that most of the phosphorylation-regulated proteins are involved in apoptosis and migration of neutrophils, which is kind of expected. Providing references to previously published studies, they tried to validate their findings, but did not do anything on their own to validate or identify a phosphosite actually being involved in migration and apoptosis. Overall, this manuscript is a descriptive one and looks like an extension of authors previously published work (Front. Mol. Biosci. 2018, 5, 89). May be this kind of analysis is a good fit for a specialized proteomics journal.
Regarding the analysis and usage of iTRAQ, there is a general disadvantage of iTRAQ technique. It can provide relative quantification only to compare relative abundances. Thereby you can only get a relative value for the concentration of an individual protein. Moreover, GPS explorer, a software package developed for iTRAQ, assumes normality in the peptide ratio for a protein once an outlier filter is applied. Some of these analytical methods try to take into account the varying precision of the peptide measurements. There are many different ways to process peptide data, but as yet no systematic study has been completed to guide iTRAQ analysis and ensure the methods being utilized are appropriate (PMID: 20382981).
Reviewer 2 Report
Review of the manuscript
The objective of this current work is to analyze the phosphoproteome of rat neutrophils in the context of intestinal ischemia and reperfusion, and the effect of ischemic preconditioning. The methodology involves induction of ischemia and reperfusion, in the presence or absence of preconditioning. Sham laparotomy and control groups are also involved. Neutrophils were isolated from blood and subjected to proteomic analysis. Analysis yielded several crucial phosphopeptides, motifs of interest, and kinases and phosphatases of interest. Several key targets are identified. The current study is hypothesis generating in its scope, and the following questions are raised:
The neutrophils are isolated from cardiac blood, followed by subsequent processing. Is this representative of the neutrophil population infiltrating the intestines after 120 minutes of reperfusion, as the experimental design suggests? It is possible that the extravasating neutrophils exposed to local inflammatory milieu undergo different phenotypic modulations, and could potentially affect the phosphoproteomic data. The comparisons among groups for phosphoproteins and kinases/phosphatases are not clear. For instance, for Grk2, the discussed differences are between the iIR and Ctrl group. For Stk38, the comparison is between iIR and iIPC. It would be helpful to compare the general implications of these changes between control and lap groups – did the impact of surgery affect the phosphoproteome? Overall, the discussion seems quite fragmented and difficult to follow. A more systematic explanation and organization is warranted. Several interesting pathways were outlined and discussed. What is the significance of insulin signaling, mTOR signaling pathway and Long-term potentiation (which is primarily a neuronal phenomenon) in the neutrophilic response post intestinal ischemia reperfusion? The discussion should further discuss the impact the influence of other cell types (intestinal epithelial cells, crypt cells and other immune cells infiltrating the intestinal mucosa) in the context of iIR, and relate it to the phosphoproteomic data obtained in this study Line 48: Should say depletion of neutrophils Line 50: Should say proteolytic enzyme (not enzymes) secretion
